# Production of an Oncolytic Adeno-Associated Virus Containing the Pro-Apoptotic TRAIL Gene Can Be Improved by shRNA Interference

**DOI:** 10.3390/ijms26020567

**Published:** 2025-01-10

**Authors:** Nicholas Donohue, Simeng Li, Stefano Boi, Alana Rainbow-Fletcher, Niall Barron

**Affiliations:** 1National Institute for Bioprocessing Research and Training, Foster Avenue, Mount Merrion, Blackrock, A94 X099 Dublin, Ireland; 2Pharmaron, 12 Estuary Banks, Speke, Liverpool L24 8RB, UK; 3School of Chemical and Bioprocess Engineering, University College Dublin, Belfield, D04 V1W8 Dublin, Ireland

**Keywords:** Adeno-associated virus, gene therapy, shRNA, oncolysis, rAAV

## Abstract

Recombinant Adeno-associated virus (rAAV) is a popular vector for treating genetic diseases caused by absent or defective genes. rAAVs can be produced that contain a therapeutic transgene, i.e., a correct copy of the affected gene, which is then delivered into target cells. A further application of rAAV is to deliver pro-apoptotic genes such as TNF-related apoptosis-inducing ligand (*TRAIL*) into cancer cells, leading to tumor regression. However, rAAV production is expensive and insufficient yields may hinder wide-spread adoption especially in systemic conditions. During rAAV production, the therapeutic transgene may be expressed in the producer cell line, and in the case of an oncolytic gene, this would likely lead to cell death thus reducing rAAV yields. Here we demonstrate that expression of TRAIL during rAAV production in HEK293F cells negatively impacts rAAV yield. A shRNA-based strategy was developed to suppress the expression of TRAIL in rAAV-producing cells specifically during the production process. Incorporating a *TRAIL*-targeting shRNA expression cassette within the backbone of the rAAV genome-encoding plasmid during triple-transfection of HEK293F cells reduced transgene expression and led to a 60% increase in the yield of rAAV-*TRAIL* compared to controls.

## 1. Introduction

Genetic diseases caused by missing or defective genes can be treated by transferring a wild-type copy of the affected gene into a patient’s cells. Because these defects may only be relevant in certain specialised cells, targeting gene therapy vectors to the correct cells is important. rAAV is one of the most popular vectors to achieve this due to its versatile range of target cells and low immunogenicity [1,2]. rAAV-based gene therapies consist of a therapeutic transgene flanked by inverted terminal repeats (ITRs) enclosed in a capsid shell, which determines the serotype and target cell range. Currently, the U.S. Food and Drug administration has approved 5 rAAV-based gene therapy drugs, with a further 136 undergoing clinical trials [3].

Besides delivering replacement genes, an alternative use for rAAV in cancer treatment is to deliver toxic or pro-apoptotic genes to tumor cells. One such transgene is TNF-related apoptosis-inducing ligand (*TRAIL*), which induces cell death by binding to the DR4 (TRAIL-R1) or DR5 (TRAIL-R2) death receptors [4,5]. rAAV-*TRAIL* has been used successfully to induce tumor regression in mouse models [6,7,8,9].

Production of rAAV-*TRAIL* (and rAAV in general) is challenging and expensive. rAAVs are commonly produced by transiently transfecting HEK293F cells with 3 plasmids encoding the ITR-flanked therapeutic transgene, the rAAV replication and capsid genes (*rep* and *cap*) and Adenovirus-derived helper genes, respectively. Following transfection, Rep binds the ITR region to initiate replication and packaging of the transgene into capsids produced from the *cap* gene [10]. While this approach is necessary to minimize the chance of recombination events leading to the production of wild-type, replication-competent rAAVs, the resulting yields are often low. Higher rAAV production yields would have positive economic implications and thus increase access to medication.

A further issue is that, unless under the control of a highly tissue-specific promoter sequence, the therapeutic transgene may be expressed in the producer cell line while the rAAV is being manufactured. This is likely to reduce yields as the cell expends resources to express the transgene instead of synthesizing rAAVs. Notably, if the transgene product is toxic to the producer cell, rAAV production may be highly compromised with a consequentially negative impact on process yield.

Tissue-specific promoters can be used to prevent undesirable transgene expression in producer cells [11]. However, ‘leaky’ expression of a toxic transgene can still occur [12,13]. Furthermore, the maximum transgene packaging capacity of rAAV is relatively small at 4.7 kb and this may limit options in promoter choice.

Several groups have reported different approaches to overcoming this challenge, including the application of RNA interference (RNAi) for gene silencing. Mediators of RNAi include microRNAs (miRNAs) and small interfering RNAs (siRNAs). The main difference is that miRNAs can regulate multiple genes, while siRNAs have a single specific target. A further category, short hairpin RNAs (shRNAs) are single-stranded RNAs with self-complementary ends; a hairpin loop at the center allows the molecule to fold back on itself into a double-stranded RNA. All types of RNAi make use of the RNA-induced silencing complex (RISC); double-stranded RNAs are processed by the endo-ribonuclease Dicer into siRNAs, then formed into a complex with RISC which degrades mRNAs complementary to the siRNA, resulting in gene silencing [14,15].

Taking advantage of the cell type-specific nature of miRNA expression, Blahetek et al. added miRNA binding sites (MBS) in the 3′ untranslated region (UTR) of the therapeutic transgene. The particular miRNA was chosen due to it being highly expressed in HEK293 cells, thus silencing the transgene mRNA during rAAV production, but not in the patient target tissue [16]. While effective, this approach does result in extra genetic material being included in the gene therapy being administered which may not be desirable. Riboswitch-mediated attenuation of transgene expression during the production stage was also shown to greatly improve the manufacturability of rAAVs containing the pro-apoptotic gene, *BAX*. By placing a guanine-responsive riboswitch sequence in the 3′ UTR of the transgene, the addition of the inducer to the production culture caused destabilization of the mRNA and thus limited Bax protein production. In this case the inclusion of the riboswitch sequence in the transgene did not interfere with expression in the target cells in mice [17].

In this study, we focused on *TRAIL* as the model oncolytic gene and show that transgene expression during rAAV production in HEK293F cells reduced cell viability and thus rAAV yields. To address this problem, a transgene-specific shRNA cassette was inserted into the backbone of the rAAV genome-encoding plasmid. Following triple transfection in HEK293F cells, shRNA expression targets the *TRAIL*-containing transcript and leads to improved rAAV production [18]. Importantly, by placing the shRNA cassette in the plasmid backbone, we avoid introducing extraneous genetic material into the resulting rAAV product.

## 2. Results

### 2.1. rAAV-TRAIL Plasmid Construction

To examine the effects of TRAIL expression during rAAV production in HEK293F cells, a *gfp*:*TRAIL* transcriptional fusion under control of the strong CMV promoter was created. The *gfp* and *TRAIL* genes were separated by an internal ribosome entry site (IRES). This cassette was inserted between the rAAV serotype 2 inverted terminal repeats in the pAAV plasmid. The backbone of this plasmid was then modified by inserting a U6 promoter-driven shRNA expression cassette, containing an anti-*gfp* shRNA sequence (Figure 1). By targeting *gfp* rather than the *TRAIL* transcript, this system could be used in future experiments to test inhibition of other toxic transgenes placed in the same cassette. A control plasmid was also generated containing a scrambled shRNA sequence. We hypothesized that expression of the anti-*gfp* shRNA would promote degradation the *gfp*:*TRAIL* mRNA via Dicer and RISC, thereby preventing TRAIL expression. In contrast, the scrambled shRNA control should not impact TRAIL expression. Directional cloning with SanDI restriction sites was used to increase the copy number of shRNA cassettes in the plasmid, with a view to increasing shRNA expression.

### 2.2. TRAIL Reduces Viability of HEK293F Cells, but shRNA Interference Alleviates This Effect

HEK293F cells were transfected with either the anti-*gfp* shRNA-containing pAAV-*gfp*-*TRAIL* or scrambled shRNA-containing pAAV-*gfp*-*TRAIL* plasmids, along with pRep2/Cap5 and pHelper. 72 h post-transfection, the viability of the HEK293F cells was measured and rAAVs were harvested and purified. Cells transfected with the anti-*gfp* shRNA plasmid showed significantly higher viability compared to the scrambled shRNA control (76.8% vs. 59.8% respectively, Figure 2a). A further control without the *TRAIL* gene or shRNA cassettes showed the highest viability (90.0%), indicating that the anti-*gfp* shRNA does not completely rescue the cells from the toxic effects of TRAIL.

### 2.3. Anti-Transgene shRNA Improves Production of rAAV-TRAIL

Harvested and purified rAAVs were quantified via qPCR and ELISA to enumerate viral genomes and capsids, respectively. The viral genome yield of samples generated using pAAV:*gfp*:*TRAIL* expressing the anti-*gfp* shRNA was significantly higher (1.6-fold increase, *p* < 0.001, *t*-test) than those containing the scrambled shRNA control (Figure 2b). rAAV yields of both the anti-*gfp* shRNA and scrambled control samples were lower than a *gfp*-only control sample indicating that even with shRNA-induced suppression, producing rAAV containing a cytotoxic transgene was not fully restored to the levels achieved when packaging a more benign cargo, i.e., *gfp*. ELISA quantification of viral capsid yields showed a similar pattern, i.e., significantly higher production in the anti-*gfp* shRNA containing sample than the scrambled control (1.7-fold increase, *p* < 0.01, *t*-test), and again, both samples were lower than the non-*TRAIL* containing control (*p* < 0.001, *t*-test) (Figure 2c). When comparing total viral genomes divided by total live cells at 72 h post transfection, significant differences were observed between the shRNA samples (8.00 × 10^3^ vg/cell and 5.79 × 10^3^ vg/cell for the anti-gfp shRNA and scramble shRNA samples, respectively; *p* < 0.05, *t*-test). However, both shRNA samples showed significantly reduced yields per cell compared to the non-TRAIL control 2.07 × 10^4^ vg/cell, *p* < 0.001, *t*-test) (Appendix A). Taken together, these results demonstrate that the addition of the anti-*gfp* shRNA improves production of rAAV-*TRAIL* without resorting to engineering the transgene sequence itself, that will be packaged in the rAAV capsid and ultimately go into the patient.

### 2.4. Plasmid Backbone-Encoded shRNA Reduces Transgene Expression in rAAV-TRAIL Producing HEK293F Cells

Flow cytometry analysis showed that expression of GFP-which acts as a surrogate for TRAIL expression as they are encoded on the same transcript-in HEK293F cells was reduced in samples containing the anti-*gfp* shRNA, compared to the scrambled control (*p* < 0.001, *t*-test). This applied to both the % of GFP positive cells as well as the mean fluorescence levels. A non-*TRAIL* control showed higher values for both criteria (*p* < 0.001, *t*-test) (Figure 3a,b). This indicated that GFP expression is reduced by both the anti-*gfp* shRNA as well as the production of TRAIL, which likely reduces cell viability and therefore the ability to express GFP.

A control experiment was performed to exclude differences in transfection efficiency between the samples causing varying levels of GFP expression. HEK293F cells were transfected with a constitutive DsRed expression plasmid, as well as pRep/Cap, pAdDelta, and pAAV-*gfp*-*TRAIL* + anti-*gfp* shRNA, or pAAV-*gfp*-*TRAIL* + scramble shRNA, or pAAV-*gfp*. DsRed expression was equal across all samples, while GFP expression differed in line with previous observations (Appendix A). This result indicated that transfection efficiency was equal across all samples and that differences in GFP were linked to shRNA and *TRAIL* expression.

### 2.5. Anti-shRNA Reduces DR5 Death Receptor Expression in rAAV-TRAIL Producing HEK293F Cells

To further investigate the mechanism of reduced yield of TRAIL-rAAVs, the expression of DR5 in transfected HEK293F cells was measured via flow cytometry. DR5 is one of the primary receptors for TRAIL and acts to trigger a pro-apoptotic pathway. DR5 is thought to be present on the surface of HEK293F cells at a low level and becomes internalized when activated by TRAIL [19,20]. Results showed that DR5 was expressed at similar levels in HEK293F cells transfected with either the anti-*gfp* shRNA or the non-*TRAIL* control. In contrast, the scrambled shRNA control showed a significantly lower percentage of DR5 positive cells than the anti-*gfp* shRNA sample (84.8% vs. 92.4%, respectively, *p* < 0.01, *t*-test), likely due to the internalization of DR5 with TRAIL expression, as well as a significantly higher mean fluorescence (*p* < 0.05, *t*-test) (Figure 3c).

### 2.6. rAAV-TRAILs Transduce CHO-K1 Cells

Finally, we assessed whether the produced rAAV-*TRAIL*s were functionally equivalent to non-*TRAIL* rAAVs in terms of transduction efficiency. Chinese hamster ovary cells (CHO-K1) were chosen as a model in vitro target cell due to reports that they are more permissive to transduction by the rAAV5 serotype compared to other cell lines [21]. CHO cells were transduced with equal amounts of rAAV-*TRAIL*s derived from both the anti-*gfp* shRNA samples and the scrambled shRNA controls as well as non-*TRAIL* containing virus. All rAAV samples were able to transduce CHO-K1 cells resulting in GFP expression. Notably, the presence of *TRAIL* did not significantly affect either expression of GFP or DR5 (Figure 4). One possible explanation for this observation is that, while more than 85% of HEK293F cells expressed DR5, less than 10% of CHO-K1 cells expressed DR5 (Figure 3c and Figure 4b). As a result, rAAV-*TRAIL*s may not have induced significant differences in GFP expression or viability compared to non-*TRAIL* control rAAVs.

## 3. Discussion

Oncolytic rAAVs are a promising new tool in cancer treatment, however production of rAAV in general is expensive based on current production yields, particularly to meet the demands of high-dose or common disease indications. This work demonstrated that expression of the cytotoxic gene, *TRAIL*, in HEK293F cells during rAAV production leads to reduced viability and increased apoptosis most likely via the DR5 receptor. Subsequently, these conditions lead to reduced yields of rAAV-*TRAIL*. A shRNA-based approach was used to inhibit *TRAIL* expression, which improved rAAV-*TRAIL* yields. Although expression of the anti-*gfp* shRNA was not sufficient to fully restore rAAV-*TRAIL* yields to the level of a non-*TRAIL* rAAV control, increasing the number of shRNA cassette repeats might offer further improvement. It might also be possible to further suppress transgene expression by using multiple shRNAs, targeting different regions of the same mRNA. However, the RNAi processing machinery of the cell may be the limiting factor here. Furthermore, optimization of the shRNA sequence could be undertaken for each individual target transgene to maximize the effect. Alternatively, expression of other elements in the apoptotic pathway could be silenced, e.g., through use of a DR5 antagonist.

GFP expression was impacted by both TRAIL and the anti-*gfp* shRNA; in the presence of TRAIL and the absence of the anti-*gfp* shRNA, cells showed reduced viability, which most likely led to reduced GFP expression. When TRAIL and the anti-*gfp* shRNA were both present, viability was restored, though not to non-TRAIL control levels. Furthermore, TRAIL-containing samples showed reduced GFP intensity in GFP-positive cells compared to the non-TRAIL control (Figure 3b). This may be the result of *gfp* being in a transcriptional fusion with *TRAIL*, which could lead to lower GFP translation. As a result, in samples with TRAIL and the anti-*gfp* shRNA, reduced viability, shRNA silencing and the transcriptional fusion with TRAIL all contributed to lower GFP expression. This would also explain the strong differences observed in the percentage of GFP positive cells, when comparing the anti-*gfp* shRNA + TRAIL sample to the non-TRAIL control (Figure 3a). A control experiment excluded the possibility of these differences being caused by varying transfection efficiency (Appendix A).

In a similar approach, expression of the toxic transgene *G protein-coupled receptor 78 (GPR78)* during AAV production was blocked by co-transfecting plasmids containing anti-*GPR78* shRNAs. AAV yields were comparable to a non-GPR78 control. However, this approach relies on transfection of a further plasmid, which would reduce the overall efficiency [22].

A further option was explored in which toxic transgene expression was reduced by inserting miRNA target sites or riboswitches in the 3′ untranslated region of the transgene [16,17]. However, inserting such sequences anywhere between the ITRs further reduces the already limited packaging capacity of rAAV (~4.7 kb). Indeed, partial packaging, where some capsids contain less than the full transgene sequence, is a recognized ‘impurity’ in existing rAAV manufacturing processes, and becomes more pronounced as the maximum sequence capacity limit is approached [23]. Additionally, using these sequences in rAAV drug production would lead to them being present in the final product administered to patients, which may not be desirable.

While rAAV-*TRAIL* successfully transduced CHO-K1 cells, no effect on DR5 expression or apoptosis was observed. CHO-K1 was chosen as the target cell line as it has been reported to be more susceptible to transduction by rAAV5, however low levels of DR5 expression mean it would potentially not be as susceptible to TRAIL signaling [21].

Overall, the system described here provides a useful in vitro screening tool to produce and test rAAVs carrying various toxic transgenes. An important benefit is that the toxic transgene can be easily exchanged without the need to re-design the anti-shRNA cassette. Once this has been established for a pro-apoptotic gene of interest, the logical next step would be to design the shRNA to target the toxic transgene directly, for the manufacturing process, and to remove the *gfp* gene.

Future applications of shRNA in rAAV manufacturing could extend to regulating other cellular pathways that interfere with rAAV production. For example, over-expression of a long non-coding RNA involved in regulating the ATP-binding cassette cellular transporter pathway resulted in increased production of rAAVs [24]. As mentioned earlier, rather than using an artificial shRNA, Blahetek et al. added artificial miRNA binding sites to the 3′ UTR of the cytotoxic transgene, which only interacted with an endogenous miRNA that is only highly expressed in HEK293 cells but not in the patient target tissue [16]. While useful in improving manufacturing characteristics, it does result in extra genetic material being included in the gene therapy going into the patient. Another option might include transient delivery of short double-stranded siRNAs targeting the cytotoxic transgene, along with the three AAV plasmids. These would suppress transgene expression in the HEK293 cells but not form any part of the AAV produced.

The approach described here is the first to use shRNAs to reduce toxic transgene expression during rAAV production, without the need for another plasmid or insertion of additional sequences that would be packaged into the final rAAV product.

## 4. Materials and Methods

### 4.1. TRAIL Vector Construction

Plasmids were acquired from Addgene (Watertown, MA, USA): pAdDeltaF6 (plasmid # 112867), pAAV2/5 (plasmid # 104964), and pAAV-*gfp* (plasmid # 32395). The *TRAIL* gene (derived from pEGFP-*TRAIL*, plasmid # 10953) was inserted downstream of *gfp* and an internal ribosome entry site. The anti-*gfp* shRNA and scramble control shRNA cassettes were inserted into the backbone of the pAAV-*gfp* plasmid, along with flanking SanDI sites. The resulting vector and PCR-amplified shRNA cassettes were digested with SanDI (Thermo Fisher Scientific, Dublin, Ireland), purified and ligated in a 1:6 vector:insert molar ratio. The resulting colonies were screened for the number of shRNA cassette inserts.

### 4.2. Cell Culture

HEK293F cells (Thermo Fisher Scientific, Waltham, MA, USA, Cat. R79007) were suspension-cultured in BalanCD HEK (Cat. 91165-1L, FUJIFILM Irvine Scientific, Wicklow, Ireland) chemically defined, serum-free medium supplemented with 4 mM L-Glutamine (Thermo Fisher Scientific, Dublin, Ireland). CHO-K1 cells were suspension cultured in CHO-SFM with anti-clumping agent (Thermo Fisher Scientific, Dublin, Ireland). All cells were cultured in 125 mL Erlenmeyer flasks (Thermo Fisher Scientific, Dublin, Ireland) at 37 °C, 5% CO_2_, 125 rpm agitation, 25 mm throw in a humidified atmosphere.

### 4.3. Triple Transfection and rAAV Purification

HEK293F cells were seeded in 4.5 mL BalanCD HEK293 + 4 mM L-Glutamine at 1.11 × 10^6^ cells/mL in 50 mL bioreactor tubes, incubated overnight at 37 °C, 8% CO_2_, 125 rpm agitation in a humidified atmosphere. On the following day, cells were triple transfected with 3 µg/mL of pAAV-*gfp-TRAIL*, pAAV2/5, and pAdDeltaF6 (molar ratio 1:1:1) and 1:2 ratio of Polyethylenimine MAX (“PEI-MAX”, Polysciences, Warrington, PA, USA). The plasmids and PEI-MAX reagent were initially diluted in two separate tubes containing 250 µL of medium. The contents of the two tubes were then mixed and incubated for 15 min at room temperature before adding to the cells. Transfections were performed in biological triplicates. For negative controls to be used in qPCR, pAAV-*gfp-TRAIL* was transfected in the same concentration used in triple transfection, but without the other plasmids. For general negative controls, samples were mock-transfected by adding 500 µL of fresh cell culture medium. 72 h post-transfection, cell viability was measured by trypan blue staining on the LUNA-II Automated Cell Counter (Logos Biosystems, Villeneuve d’Ascq, France). 4 mL of each culture were collected and centrifuged at 16,000× *g*, 20 min, 4 °C. The supernatants were aliquoted and stored at −80 °C for further analysis, while the cell pellets were resuspended in 1 mL of medium. Cell pellets were processed by 3 liquid nitrogen/37 °C water bath freeze/thaw cycles. After the final thaw, MgCl_2_ and Benzonase (Merck, Darmstadt, Germany) were added to the samples to a final concentration of 2 mM and 100 U/mL, respectively. Samples were incubated at 37 °C for 1 h. Following incubation, samples were centrifuged at 16,000× *g*, 20 min, 4 °C. Supernatants were aspirated and stored at −80 °C until further analysis.

### 4.4. qPCR

rAAV viral genome titers were determined for supernatant and cell pellet samples by qPCR. First, 10 µL of each sample was treated with 1.25 U/µL Benzonase (Merck, Darmstadt, Germany), followed by 6 U/mL proteinase K (Thermo Fisher Scientific, Dublin, Ireland) digestion. Two dilutions of the digested sample (1:100 and 1:1000) were used for qPCR, performed using Fast SYBR Green Master Mix (Applied Biosystems, Carlsbad, CA, USA) and primers specific to the rAAV2 ITRs flanking the *gfp* expression cassette in the pAAV-*gfp-TRAIL* plasmid. qPCR was performed using a QuantStudio 3 Real-Time PCR System (Thermo Fisher Scientific, Dublin, Ireland), and the viral genome titers were calculated using the QuantStudio design and analysis software (version 1.5.1) based on the generated standard curve.

### 4.5. ELISA

ELISA measurements were performed using a rAAV5 kit (Progen, Heidelberg, Germany), according to the manufacturer’s instructions. Results were obtained using a Biotek Synergy H1 plate reader at 450 and 650 nm wavelengths (Agilent, Santa Clara, CA, USA).

### 4.6. Transduction

For transduction experiments, 1 × 10^5^ CHO-K1 cells were seeded into each well of a 24-well plate and grown in DMEM-F12 + 10% fetal bovine serum (Thermo Fisher Scientific, Dublin, Ireland) for 24 h, before adding 22,000 vg/cell, then incubating a further 72 h in a static incubator before harvesting. To recover cells which were attached to the culture ware surface, 200 µL of 0.25% trypsin-EDTA (Thermo Fisher Scientific, Dublin, Ireland) was added into each well and incubated for 3 min at 37 °C. Cells recovered in Trypsin-EDTA were combined with harvested suspension cells. After washing with phosphate buffered saline (Thermo Fisher Scientific, Dublin, Ireland), cells were stained and examined by flow cytometry.

### 4.7. Flow Cytometry

Samples were fixed in 1 mL of 4% *v*/*v* formaldehyde/Phosphate buffered saline (PBS) solution for 15 min, then centrifuged at 300× *g*, 5 min, 20 °C. The supernatant was discarded, and the cell pellet resuspended in 1 mL PBS. This wash step was repeated two times. Samples were loaded onto a 96-well U-bottom plate and analyzed by FACS Accuri C6 (BD, Franklin Lakes, NJ, USA). For staining with PE anti-DR5 antibody (Abcam, Cambridge, U.K., RRID: AB_879820), 1 mL of cells were harvested, centrifuged at 300× *g*, 5 min, 20 °C and resuspended in ice-cold PBS, 10% Fetal calf serum, 0.1% sodium azide solution (Thermo Fisher Scientific, Dublin, Ireland). 2 µL of antibody was added to the cells and incubated for 30 min at room temperature in the dark. Following centrifugation at 300× *g*, 5 min, 20 °C, cells were resuspended in 100 µL 4% formaldehyde and incubated at room temperature for 15 min. Finally, centrifugation was performed at 300× *g*, 5 min, 20 °C and cells were resuspended in 1 mL ice-cold PBS. The final wash step was repeated twice, followed by analysis using FACS Accuri C6 as described above.

### 4.8. Statistical Analysis

Statistical analysis was performed using Prism (Graphpad, Boston, MA, USA) and Student’s two sample unpaired *t*-test with a significance level *p* < 0.05.

## Figures and Tables

**Figure 1 ijms-26-00567-f001:**
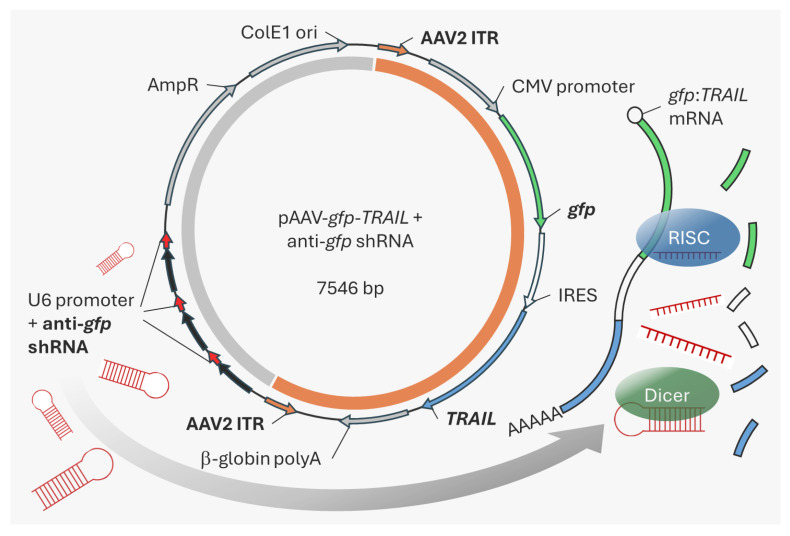
Plasmid design of pAAV-*gfp*-*TRAIL* with anti-*gfp* shRNA cassettes. The rAAV transgene region of the plasmid (bordered by rAAV2 ITR regions, orange arrows) is indicated by the orange semi-circle. The remaining plasmid backbone containing the anti-*gfp* shRNA cassettes is marked in grey. Anti-*gfp* shRNAs (red arrows) expressed under control of the U6 promoter (black arrows) target *gfp*:*TRAIL* (green, white and blue arrows) transcripts for Dicer/RISC mediated degradation.

**Figure 2 ijms-26-00567-f002:**
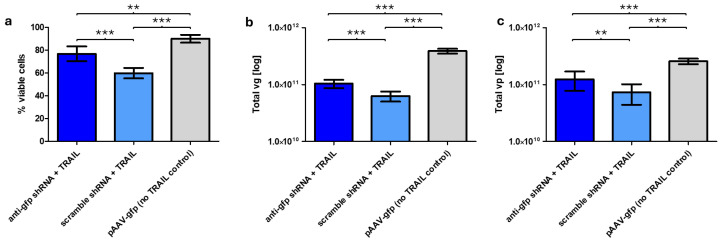
HEK293F cells, 72 h post triple-transfection with pRep/Cap, pAdDelta and pAAV-*gfp*-*TRAIL* + anti-*gfp* shRNA OR pAAV-*gfp*-*TRAIL* + scramble shRNA OR pAAV-*gfp* (no *TRAIL* control). (**a**) Viability measured by Luna cell counter. Each bar represents 6 biological repeats (=6 separate transfections) for the *TRAIL* samples. pAAV-*gfp* (no *TRAIL* control) bar represents 4 biological repeats. (**b**) qPCR quantification of total viral genome (vg) yield (log-scale), harvested at 72 h post-transfection. Each bar represents 6 biological repeats (=6 separate transfections) and 12 technical repeats for the *TRAIL* samples. pAAV-*gfp* (no *TRAIL* control) bar represents 4 biological repeats and 8 technical repeats. (**c**) ELISA data showing total viral particle (vp) yield (log-scale), harvested at 72 h post-transfection. Each bar represents 6 biological repeats (=6 separate transfections) and 12 technical repeats for the *TRAIL* samples. pAAV-*gfp* (no *TRAIL* control) bar represents 4 biological repeats and 8 technical repeats. Error bars represent ± SD, ** = *p* < 0.01, *** = *p* < 0.001 (*t*-test).

**Figure 3 ijms-26-00567-f003:**
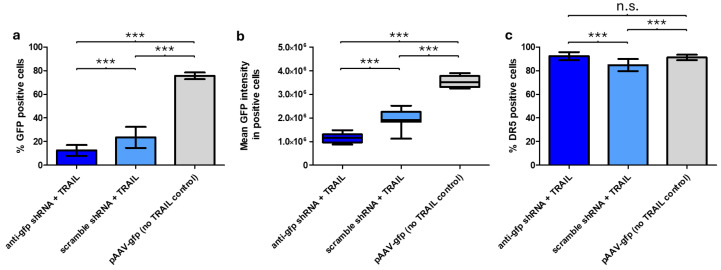
HEK293F cells, 72 h post triple-transfection with pRep/Cap, pAdDelta and pAAV-*gfp*-*TRAIL* + anti-*gfp* shRNA OR pAAV-*gfp*-*TRAIL* + scramble shRNA OR pAAV-*gfp* (no *TRAIL* control). Cells were incubated with anti-DR5 antibody (PE labelled), then examined by FACS Accuri C6. (**a**) % GFP positive cells. (**b**) Mean GFP intensity for GFP positive cells. (**c**) % DR5 positive cells. Bars/boxes represent a minimum of 4 separate transfections. Error bars represent ± SD. *** = *p* < 0.001, n.s., not significant, *t*-test.

**Figure 4 ijms-26-00567-f004:**
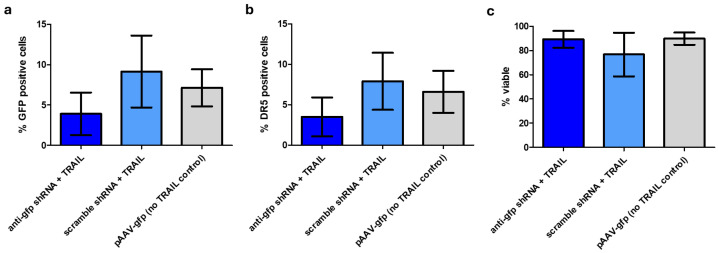
CHO-K1 cells transduced with rAAV-*TRAIL*s, measured after 72 h, examined by FACS Accuri C6. (**a**) % GFP positive. (**b**) % DR5 positive. (**c**) % of viable cells. Bars represent 3 separate transductions. Error bars represent ± SD. No significant differences observed.

## Data Availability

The original contributions presented in this study are included in the article/Appendix A. Further inquiries can be directed to the corresponding author.

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
