# Peer review of "Production of an Oncolytic Adeno-Associated Virus Containing the Pro-Apoptotic TRAIL Gene Can Be Improved by shRNA Interference"

_ijms, 2025, doi:10.3390/ijms26020567_

Round 1

Reviewer 1 Report

Comments and Suggestions for Authors

Peer Review Summary

The manuscript presents a strategy to enhance host cell performance and increase recombinant AAV (rAAV) yield in a triple plasmid transient transfection process through shRNA interference to suppress “toxic” transgene expression. While the study demonstrates some novelty and provides supporting data for its conclusions, I recommend acceptance only after addressing the following major concerns:

Major Concerns:

1.     Introduction: Additional background information and details on the design strategies related to shRNA are needed, as shRNA technology may be less familiar in the context of industrial rAAV manufacturing.

2.     Broader Applicability: Although the study includes adequate biological replicates, the broader applicability to rAAV manufacturing is not fully supported by the limited data. The study would benefit from expanded experiments across various serotypes (e.g., AAV2 and AAV8) and production cell lines (e.g., HEK293, 293T, and HeLa) to reinforce its conclusions.

3.     Transduction Experiment: The transduction experiment would be more convincing if additional cell lines, such as HEK293, were included to assess GFP and DR5 expression. Currently, CHO cells are used due to their compatibility with AAV5; however, the absence of TRAILs did not affect GFP or DR5 expression while the author give some explanations.

4.     Transfection Efficiency: Data on transfection efficiency should be provided to ensure consistency across transfection conditions (especially in the scramble shRNA group and the no TRAIL control group). This data is essential to comprehensively assess rAAV productivity.

5.     Cell Growth and Productivity Data: Adding viable cell density (VCD) post transfection and specific productivity (Qp) data would further clarify the factors contributing to the observed decrease in rAAV productivity in shRNA and scramble shRNA groups.

6.     Flow Cytometry Analysis: Further discussion is required to address the notable difference in GFP-positive cell percentages between the scramble shRNA group (~20%) and the no TRAIL control (~80%). Additionally, transfection efficiency data for the three groups should be presented, with further discussion based on these observations.

7.     shRNA Strategy Discussion: A more in-depth discussion of the challenges, advancements, and future perspectives of shRNA technology for rAAV manufacturing is needed to support its potential in the biomanufacturing field.

Minor Concerns:

  • Line 220: Additional clarification is needed for “IRES – still intact???”

Author Response

Major Concerns:

  1. Introduction: Additional background information and details on the design strategies related to shRNA are needed, as shRNA technology may be less familiar in the context of industrial rAAV manufacturing.

 We thank the reviewer for this suggestion. We have expanded the introduction to include further background information on shRNA technology (page 2, lines 64-73).

  1. Broader Applicability: Although the study includes adequate biological replicates, the broader applicability to rAAV manufacturing is not fully supported by the limited data. The study would benefit from expanded experiments across various serotypes (e.g., AAV2 and AAV8) and production cell lines (e.g., HEK293, 293T, and HeLa) to reinforce its conclusions.

The presented data is intended as a proof-of-concept for industrial rAAV production. For this reason, we chose AAV5 as our model, as this serotype is one of the most common serotypes used in rAAV-based drugs –of the 5 currently FDA-approved rAAV drugs, 2 are rAAV5, while the remaining 3 are all different serotypes.

HEK293 is the most common cell line used for manufacturing AAVs. While HeLa can also be used, this is much less common. Notably, HeLa cells require the use of a helper virus (e.g. Adenovirus) to produce AAV1. HeLa is therefore not ideal for industrial AAV production.

HEK293T (suggested in the reviewer’s comment) and HEK293F are both derived from the same original HEK293 cell line. However, HEK293T contains the Simian Virus T40 antigen, which raises safety concerns in manufacturing2. In our opinion, HEK293T is unlikely to be vastly different from HEK293F in terms of rAAV production and less relevant in an industrial setting.

1Weiheng Su, Leonard W. Seymour & Ryan Cawood. AAV production in stable packaging cells requires expression of adenovirus 22/33K protein to allow episomal amplification of integrated rep/cap genes. Scientific Reports, volume 13, Article number: 21670 (2023)

2Vilchez, R. A., Kozinetz, C. A., Arrington, A. S., Madden, C. R., & Butel, J. S. (2003). Simian virus 40 in human cancers. The American Journal of Medicine, 114(8), 675– 684. https://doi.org/10.1016/s0002-9343(03)00087-1

  1. Transduction Experiment: The transduction experiment would be more convincing if additional cell lines, such as HEK293, were included to assess GFP and DR5 expression. Currently, CHO cells are used due to their compatibility with AAV5; however, the absence of TRAILs did not affect GFP or DR5 expression while the author give some explanations.

Ellis et al. (2013, Reference 21 in the text) performed transduction experiments using different AAV serotypes and a range of cell lines and primary cells. For AAV5, CHO showed 9% GFP positive cells after transduction with 100,000 viral genomes per cell. However, transduction of HEK293 cells using the same concentration produced only 1% of GFP positive cells. Therefore, HEK293 is probably unsuitable for this experiment, as it would be difficult to detect any transduction at all. In fact, in that study, CHO had the highest transduction efficiency of a large panel of cell lines tested, which was the reason we chose it.

  1. Transfection Efficiency: Data on transfection efficiency should be provided to ensure consistency across transfection conditions (especially in the scramble shRNA group and the no TRAIL control group). This data is essential to comprehensively assess rAAV productivity.

Transfection efficiency was assessed by performing quadruple transfection of all three rAAV plasmids along with a plasmid expressing DsRed. While GFP expression showed the same pattern as observed in Figure 3a and Figure 3b, DsRed expression was equal among all samples (Figure S2 and page 5, lines 178-185).  This result indicates that transfection efficiency was consistent.

  1. Cell Growth and Productivity Data: Adding viable cell density (VCD) post transfection and specific productivity (Qp) data would further clarify the factors contributing to the observed decrease in rAAV productivity in shRNA and scramble shRNA groups.

Figure 2a shows the percentage of viable cells at 72 hrs post transfection.

Measuring specific productivity would require collection of rAAV at two different time-points. However, during harvesting, all cells are lysed to release the rAAV contained within. While it would be possible to take samples of the culture’s supernatant and quantify rAAV in this, the majority of produced rAAVs are contained within the cells. As a result, the actual rAAV production cannot be measured without destroying the cells, which in turn necessitates the use of a single harvesting time-point.

However, as an approximation of specific productivity we have now included data on the viral genome production per cell (page 4, lines 143-148, and Figure S1).

  1. Flow Cytometry Analysis: Further discussion is required to address the notable difference in GFP-positive cell percentages between the scramble shRNA group (~20%) and the no TRAIL control (~80%). Additionally, transfection efficiency data for the three groups should be presented, with further discussion based on these observations.

We have updated the text by expanding the discussion on the differences in GFP expression (page 6, lines 247-259). Transfection efficiency data is provided in Figure S2.

In terms of transfection efficiency, Figure 3a provides data on %GFP positive cells at 72 hrs post-transfection. Figure 3b shows data on mean GFP intensity in positive cells at the same time-point.

Our intention was to reduce expression of TRAIL as well as the co-transcribed GFP reporter. As a result, the sample containing TRAIL and the anti-gfp shRNA showed the least amount of GFP expression.

The reason for the high GFP signal in the no TRAIL control compared to the other samples is likely due to the fact that the transgene is quite different. In the latter, there is an IRES downstream of the GFP ORF, leading into the TRAIL ORF – which are transcribed as one long transcript. The translation efficiency of the GFP ORF is likely to be less efficient than the GFP-only control which is followed directly by a polyA signal.  

  1. shRNA Strategy Discussion: A more in-depth discussion of the challenges, advancements, and future perspectives of shRNA technology for rAAV manufacturing is needed to support its potential in the biomanufacturing field.

This has now been added (page 7, lines 284-295).

Minor Concerns:

  • Line 220: Additional clarification is needed for “IRES – still intact???”

This has been corrected.

Reviewer 2 Report

Comments and Suggestions for Authors

This paper demonstrated that expression of the cytotoxic gene, TRAIL, in HEK293F cells during recombinant adeno-associated virus (rAAV) production leads to reduced viability and increased apoptosis most likely via the DR5 receptor. Subsequently, these conditions lead to reduced yields of rAAV-TRAIL. To improve this rAAV production, a shRNA-based approach was used to inhibit TRAIL expression, and the results showed 1.6-fold increase in rAAV-TRAIL yields. There are some concerns as listed in the following:

*L26. 139: HEK293 cells -> HEK293F cells

**L184: delete * = p < 0.05, ** = p < 0.01, because no such markers in the Fig. 3

**L220: IRES – still intact???

L249: pEGFP-TRAIL, (plasmid # 10953)

L263, 268: 8% CO2 -> CO2 (2, subscript)

*L321, 324, 326, 327, 328,329: mins -> mins vs. min (L315), check all.

L335: give significant level p < 0.05?

**L351: References: No page number

Ref. 3: Front Med (Lausanne). 2022 Feb 9:8:809118. doi: 10.3389/fmed.2021.809118. eCollection 2021.

Ref. 14: Mol Ther Methods Clin Dev. 2024 Jun 10;32(3):101280. doi: 10.1016/j.omtm.2024.101280. eCollection 2024 Sep 12.

Ref. 20: Mol Ther Methods Clin Dev. 2024 Jan 17;32(1):101188. doi: 10.1016/j.omtm.2024.101188. eCollection 2024 Mar 14.

Author Response

We thank the reviewer for these helpful comments and have made corrections as indicated below:

*L26. 139: HEK293 cells -> HEK293F cells

Changed throughout the text.

**L184: delete * = p < 0.05, ** = p < 0.01, because no such markers in the Figure 3

Corrected, page 4, line 164

**L220: IRES – still intact???

Deleted

L249: pEGFP-TRAIL, (plasmid # 10953)

Page 7, line 306

L263, 268: 8% CO2 -> CO(2, subscript)

Page 8, line 320

*L321, 324, 326, 327, 328,329: mins -> mins vs. min (L315), check all.

Changed throughout the text.

L335: give significant level p < 0.05?

Page 9, line 393

**L351: References: No page number

Ref. 3: Front Med (Lausanne). 2022 Feb 9:8:809118. doi: 10.3389/fmed.2021.809118. eCollection 2021.

Page 9, line 417

Ref. 14: Mol Ther Methods Clin Dev. 2024 Jun 10;32(3):101280. doi: 10.1016/j.omtm.2024.101280. eCollection 2024 Sep 12.

Page 10, lines 449-451

Ref. 20: Mol Ther Methods Clin Dev. 2024 Jan 17;32(1):101188. doi: 10.1016/j.omtm.2024.101188. eCollection 2024 Mar 14.

Page 10, lines 466-468

Reviewer 3 Report

Comments and Suggestions for Authors

In this manuscript, the authors present their solution to improving the production of rAAVs delivering toxic or pro-apoptotic genes – as those are expressed in the producer cell line while rAAV manufacturing, the rAAV yield is negatively affected. As a solution, they insert a transgenes-specific shRNA cassette into the backbone of rAAV genome-encoding plasmid. In theire experiments, TRAIL was the model oncolytic gene. Specifically, TRAIL-GFP fusion enabled by an IRES sequence was used, and an anti-GFP shRNA sequence, which also allows targeting of the lethal genes in a generic way, was used to promote the degradation of TRAIL-GFP mRNA transcript. The viability of transfected HEK293 cells, which express DR5 TRAIL ligand, was improved when anti-gfp shRNA was used, but not when scrambled shRNA was used, and the production of rAAV-TRAIL was indeed enhanced. Then the system was also tested in CHO-K1 cells, where only a low percentage (5-10%) expressed DR5, and here the cells were transduced either with rAAV-TRAILs produced with the anti-GFP shRNA samples and the scrambled shRNA controls as well as the virus not containing TRAIL. Either the expression of DR5 or GFP, as well as the viability, were affected by TRAIL in comparison with non-TRAIL AAVs. The presented method is very elegant as it does not require additional plasmid for transfection, and the novel sequences are also not packaged into rAAV product.

The manuscript is clearly written, the data is well organized and well presented. Please find below a list of remarks which I hope you will find helpful.

-          For Discussion: Figure 2a shows that the viability of host cells using anti-GFP shRNA in improved, but not completely restored: are there measures to further improve the system?  

-           

-          It would be interesting to compare the findings from the reference: doi: 10.1089/hum.2019.249 (another approach by improving AAV production with shRNA)

-          Line 220: IRES – still intact??? – this sentence is probably incomplete?

-          Line 263 and throughout the text: CO2, 2 into subscript

-          Line 310: 10exp5

-          Line 321: please list all centrifugation steps with revolutions, time and temperature

-          Line 324: all antibodies should be listed with RRIDs

Author Response

-          For Discussion: Figure 2a shows that the viability of host cells using anti-GFP shRNA in improved, but not completely restored: are there measures to further improve the system?  

Many thanks for this comment! As mentioned in the discussion (page 6, lines 238-246), it would certainly be possible to add further copies of the shRNA cassette to the plasmid backbone. However, it is likely that the cells’ RNA processing machinery would eventually become the limiting factor here.

Further work could incorporate alternative methods of transgene silencing, e.g. by silencing other elements of the apoptotic pathway. Use of a DR5 antagonist may also be worth exploring. The discussion has been expanded to include this (page 6, lines 245-246).

-          It would be interesting to compare the findings from the reference: doi: 10.1089/hum.2019.249 (another approach by improving AAV production with shRNA)

The discussion has been expanded to include this reference (pages 6-7, lines 260-264).

-          Line 220: IRES – still intact??? – this sentence is probably incomplete?

Deleted

-          Line 263 and throughout the text: CO2, 2 into subscript

Page 8, line 320 and changed throughout the text.

-          Line 310: 10exp5

Page 8, line 367

-          Line 321: please list all centrifugation steps with revolutions, time and temperature

Page 8, line 337 and throughout the text

-          Line 324: all antibodies should be listed with RRIDs

Page 9, line 381